# An Adaptive Hybrid Sampling Method for Free-Form Surfaces Based on Geodesic Distance

**DOI:** 10.3390/s23063224

**Published:** 2023-03-17

**Authors:** Chen Chen, Huakun Jia, Yang Lu, Xiaodong Zhang, Haohan Chen, Liandong Yu

**Affiliations:** College of Control Science and Engineering, China University of Petroleum (East China), Qingdao 266580, China

**Keywords:** adaptive sampling, free-form surface, non-uniform rational B-spline (NURBS), geodesic distance

## Abstract

High precision geometric measurement of free-form surfaces has become the key to high-performance manufacturing in the manufacturing industry. By designing a reasonable sampling plan, the economic measurement of free-form surfaces can be realized. This paper proposes an adaptive hybrid sampling method for free-form surfaces based on geodesic distance. The free-form surfaces are divided into segments, and the sum of the geodesic distance of each surface segment is taken as the global fluctuation index of free-form surfaces. The number and location of the sampling points for each free-form surface segment are reasonably distributed. Compared with the common methods, this method can significantly reduce the reconstruction error under the same sampling points. This method overcomes the shortcomings of the current commonly used method of taking curvature as the local fluctuation index of free-form surfaces, and provides a new perspective for the adaptive sampling of free-form surfaces.

## 1. Introduction

Free-form surface parts are widely used in aerospace, automotive and other high-end equipment manufacturing fields [1]. It is one of the hotspots in the measurement field to measure their precise geometric quantities and ensure that they meet the accuracy requirements. This kind of part has a complex structure and large size, which also puts forward higher requirements for the dexterity and space of measuring equipment. The geometric measurement of a free-form surface can adopt a contact or non-contact measurement [2]. In the process of the geometric measurement of a free-form surface, the geometric error can be evaluated from two aspects: measurement accuracy and measurement efficiency. The measurement accuracy can be quantified by measuring the deviation between the actual value and the real value. When the measurement accuracy meets the measurement requirements, the measurement efficiency can be improved by optimizing the number and location of the measurement points and the measurement path [3,4,5,6].

As shown in Figure 1a, the coordinate measuring machine (CMM) equipped with a contact probe is one of the common contact measuring instruments [7]. It can obtain the point coordinates of the parts to be measured by scanning the measurement point by point. The point coordinates of the parts to be measured are reconstructed into free-form surfaces, and the geometric errors are compared with the original surfaces to accurately evaluate the machining errors generated in the manufacturing process of free-form surface parts. As shown in Figure 1b, the articulated industrial robot equipped with an optical probe is one of the common non-contact measuring instruments. The scanning measurement path is generated based on the measurement points of the parts to be measured with significant geometric characteristics, and the industrial robot drives the optical probe to move along the scanning measurement path. The scanning point cloud is reconstructed into a free surface, and the geometric error is compared with the original surface to determine the machining error of the free surface [8].

When measuring the geometric quantity of the parts to be measured, the number of measuring points is positively related to the measurement accuracy. The more the number of measuring points, the higher the measurement accuracy will be. When the number of measuring points reaches a certain value, the measurement accuracy will not change significantly. However, if the number of measuring points is too large, the measurement time will be significantly increased and the measurement efficiency will be reduced. Therefore, when measuring the components of the free-form surface to be measured, the number and position of the sampling points on the free-form surface should be reasonably distributed to ensure that the measurement efficiency can be improved as much as possible on the premise of meeting the measurement accuracy.

By setting a reasonable sampling plan, the corresponding measurement path is generated, and the free surface is measured using a contact measurement or non-contact measurement. Thus, the high-precision measurement of free-form surfaces can be realized, and the measurement efficiency can be significantly improved.

Blind sampling is a sampling method that does not consider the geometric characteristics of the free-form surface [9]. The number and location of sample points determine the geometric error of the reconstructed free-form surface. Uniform sampling is a simple and efficient sampling method, which is widely used in the field of free-form surface sampling. Dunbar et al. [10,11] introduced a random sampling strategy of fast disk under arbitrary dimension, which can realize blind sampling of a free-form surface simply and efficiently. Woo et al. [12] sampled the surface to be measured using the Hammersley distribution method. The research results show that blind sampling has the characteristics of simple and efficient sampling, but it cannot adaptively change the number and position of the sampling points based on the geometric characteristics of the free-form surface to be measured, thus affecting the measurement accuracy and efficiency.

Adaptive sampling generates sampling points according to the geometric characteristics of a free-form surface. In short, more sampling points are generated in the area with a large fluctuation of the free-form surface, and fewer sampling points are generated in the area with a small fluctuation of the free-form surface [13]. Ren et al. [14] proposed a method of using the curvature change matrix of adjacent points as the change index of a free-form surface. According to the proposed index, the optimal position of the newly added bi-directional curve mesh was determined, and the free-form surface was reconstructed based on the Gordon surface fitting principle. Javad et al. [15] used the optimization method and particle swarm optimization algorithm to optimize the position of the sampling points. On the basis of the initial sampling points, the optimal position of the new sampling points was determined by iterative optimization. When the geometric error between the reconstructed surface and the original surface or the number of sampling points reached the set threshold, the optimization was terminated. Gao et al. [16] took the aeroengine blade as the measurement object, and adaptively generated the sampling points based on the bending moment theory for the regions with different curvatures of the engine blade, so that the number and position of the sampling points could be accurately determined according to the fluctuation of the blade surface. In consideration of the influence of Gaussian curvature on machining error, Sang et al. [17] proposed a scanning line distribution strategy based on star pattern, which classifies peak points and anchor points, and connects the error peak points in different regions and anchor points in the same region, so that more scanning lines are generated in the region with small machining error, and fewer scanning lines are generated in the region with small machining error. Jiang et al. [18] studied a calculation method based on curve chord deviation, and adopted a two-step sampling method. First, based on the radius of the CMM probe ball as the threshold, adaptive Isoparametric sampling was carried out on the leading edge curve and trailing edge surface of the blade. Secondly, adaptive sampling points were selected on the Isoparametric based on the proposed curve chord deviation theory, and finally, the adaptive sampling of the blade was realized. Suleiman et al. [19] proposed a patch based free-form surface adaptive sampling method, which sorts patches according to the Gaussian curvature of each patch, determines the number of sampling points of each patch according to the sorting size, and selects the points of maximum curvature, minimum curvature and average curvature in the patch as the sampling points. If the number of sampling points exceeds these three types of points, the maximum curvature, minimum curvature and average curvature are taken as the sampling points, and so on until the number of sampling points reaches the threshold of the number of sampling points. Mansour [20] studied an adaptive sampling method for reducing the number of measurement points and improving the measurement efficiency with the blade as the measurement object. Based on the least square method, the minimum number of points required for the curve polynomial is found, so that the fitting error expressed by the polynomial curve and cubic curve is minimal. He et al. [21] proposed an adaptive sampling method for a free-form surface based on the machining error model. The machining error model was established based on the curvature of the free-form surface, and the relationship between Gaussian curvature and machining error was obtained. The adaptive sampling of a free-form surface was carried out based on the error model and the Hammersley principle. The number of sampling points was large in the places with large errors and small in the places with small errors. Yi et al. [22] discretized the triangular mesh of a free surface, and simplified the triangular mesh by iteratively shrinking the triangle edges. Based on the principle of minimum quadratic error, the optimal objective vertex under discrete curvature constraint is determined. By limiting the side length of the triangular mesh to control the number of sampling points, the adaptive sampling of free-form surfaces is realized. Yu et al. [23] selected the initial sampling point set, reconstructed the initial point set and solved it with the original surface to obtain the global error. The point with the largest global error is added to the initial point set to obtain the updated initial point set. The reconstructed surface and the original surface are solved to obtain the global error. The point with the maximum global error between the reconstructed surface and the original surface is added to the initial point set as a new sampling point. Cycle in turn until the global error reaches the set precision threshold or the number of sampling points reaches the set number threshold. Mian et al. [24] studied the influence of different sampling strategies on surface reconstruction accuracy, and the influence of workpiece size and machining quality on sampling methods. Gohari et al. [25] used principal component analysis to dynamically generate sampling points, thus reducing the number of sampling points, reducing the measurement cost and improving the measurement efficiency.

For the precise measurement of geometric quantities of free-form surfaces, the number of samples and the position of sampling points should be reasonably set to improve the measurement accuracy and efficiency. In this study, geodesic distance is used as the index of global fluctuation of a free-form surface. The free-form surface is divided into blocks, and the number of samples is adaptively determined according to the changes in the geometric characteristics of the free-form surface. Combined with the Isoparametric distribution, Poisson distribution, Hammersley distribution and NRook distribution, the distribution location of the sampling points is determined. Therefore, an adaptive hybrid sampling method for a free-form surface based on geodesic distance is proposed, which can effectively improve the measurement accuracy and efficiency.

The rest of this paper is organized as follows: Section 2 introduces the free-form surface modeling method; Section 3 introduces the definition of geodesic distance and its solution method in detail, including three sub-steps; in Section 4, the number of sampling points and the distribution of the sampling points are given; in Section 5, the reconstruction errors between the reconstructed surface and the original surface are solved and the results are analyzed; Section 6 summarizes the conclusions and future work prospects.

## 2. Free-Form Surface Modeling

Non uniform rational B-spline (NURBS) is one of the most commonly used parametric mathematical models in free-form surface geometric modeling. NURBS surfaces are widely used in the field of computer aided geometric design (CAGD), and are widely used in the geometric representation of complex components in aerospace, automobile and other fields.

The free-form surface can be represented by the control points and degrees in the u and v directions. The local adjustment of the free-form surface can be realized by adjusting the control points and weight coefficients. The NURBS surface is usually obtained by using the tensor product of two NURBS curves with two independent parameters, u and v.

The expression of a NURBS surface is as follows [26].
(1)$$S\left(u,\;v\right)=\frac{\sum_{i=0}^{m}\sum_{j=0}^{n}w_{i,j}N_{i,p}\left(u\right)N_{j,q}\left(v\right)P_{i,j}}{\sum_{i=0}^{m}\sum_{j=0}^{n}w_{i,j}N_{i,p}\left(u\right)N_{j,q}\left(v\right)}$$
where $P_{i,j}$ is the control point; m is the number of control points in the u direction; n is the number of control points in the v direction; p is the degree of the parameter coordinate u; q is the degree of the parameter coordinate v; $N_{i,p}\left(u\right)$ is a basis function of order p; $N_{i,p}\left(v\right)$ is a basis function of order q; U and V are defined knot vectors, $U=\left\{0,\cdots ,0,u_{p+1},\cdots ,u_{r-p-1},1,\cdots ,1\right\}$, $V=\left\{0,\cdots ,0,v_{q+1},\cdots ,v_{s-q-1},1,\cdots ,1\right\}$, which specify the distribution of parameters u and v; $w_{i,j}$ is the same as the $P_{i,j}$ relevant weight coefficient. The basis function is recursively defined by the Cox–deBoor algorithm, where $N_{i,p}\left(u\right)$:(2)$$N_{i,1}=\left\{\begin{array}{l} 1\;\mathrm{if}\;u_{i}<u<u_{i+1}\\ 0\;\mathrm{otherwise} \end{array}\right.$$
(3)$$N_{i,\;p}\left(u\right)=\frac{u-u_{i}}{u_{i+p-1}-u_{i}}N_{i,p-1}\left(u\right)+\frac{u_{i+p}-u}{u_{i+p}-u_{i+1}}N_{i+1,p-1}\left(u\right)$$

The NURBS surface is shown in Figure 2. The blue curve is the isoparm in the u direction, the red curve is the isoparm in the v direction, and the red surface and the blue curve form are the isoparm mesh of the NURBS surface.

## 3. Geodesic Distance

On the Riemannian manifold, the geodesic is defined as the shortest path between the points on the model surface. Geodesic distance is the distance value of geodesic. Compared with the curvature, the advantage of geodesic is that it can describe the free-form surface globally, and the curvature can only be described based on the local part of a free-form surface [27]. Therefore, the geodesic distance is used as the index of the surface fluctuation of the free-form surface in this paper, which can accurately describe the global fluctuation of the free-form surface and overcome the limitation that curvature can only describe the local fluctuation.

The geodesic distance solution method used in this paper is the thermodynamic method proposed by Crane [28]. It can be imagined that a hot needle touches a point x on the surface, which is a hot core point. With the passage of time, the heat on this point x diffuses to the rest of the surface. The heat at point y on the surface can be expressed by a thermal kernel function $\;k_{t,x}\left(y\right)\;$ to describe how the geodesic distance between any point x, y on the Riemannian manifold *φ*$\left(x,\;y\right)$ can be recovered by the point state transformation of the thermal core, as shown in Formula (4). The geodesic distance can be recovered by solving the direction of thermal motion [29].
(4)$$\varphi \left(x,y\right)=\underset{t\to 0}{\lim}\sqrt{-4t\log k_{t,x}\left(y\right)}$$

Step 1: Solve the thermal kinematics in Equation (5) by describing the propagation state of heat, and establish the temperature scalar field U. The time dispersion of the thermal propagation equation is
(5)$$\frac{\partial u}{\partial t}=∆u$$

The thermal kinematics equation is discretized and sorted to obtain
(6)$$\left(id-t∆\right)u_{t}=u_{0}$$
where id is the identity matrix, t is the time interval, ∆ is the discrete Laplacian operator, $u_{t}$ is the thermal state at time *t*, and $u_{0}$ is the thermal state at the initial time.

Step 2: The thermal gradient direction calculated in step 1 is the same as the gradient direction of the geodesic distance. Since the gradient of the geodesic distance is a unit vector, the gradient of the geodesic distance is obtained by normalizing the thermal gradient direction.
(7)$$\;X=-\nabla u/\left|\nabla u\right|$$
where X is the gradient of the geodesic distance.

Step 3: after the gradient of geodesic distance is obtained through step 2, the geodesic distance is solved through Formula (8):(8)$$\underset{\phi }{\min}\int_{M}{\left|\nabla \phi -X\right|}^{2}$$

According to the variational method, the minimization of Formula (8) is the Poisson equation.
(9)Δϕ=∇⋅X
where ϕ is the geodesic distance between the vertex and the hot core point.

Based on the principle of the NURBS surface in part 2 of this paper, the free-form surface to be measured is shown in Figure 3, with a size of 50 mm × 50 mm. Reduce the dimension of the free-form surface from the three-dimensional space to the two-dimensional parameter domain $\left(u,v\right)$; carry out Isoparametric sampling on the two-dimensional parameter domain $\left(u,v\right)$; map the sampling points on the two-dimensional parameter domain to the three-dimensional space; use the collection of sampling points on the three-dimensional space to replace the free-form surface, and set the number of sampling points to (41 × 41) = 1681.

In order to evaluate the global fluctuation of a free-form surface, it is necessary to segment the surface. The principle of the surface segment is shown in Figure 4 and Figure 5. As shown in Figure 4, the free-form surface to be measured is divided into four surface segments $P_{i}$ in the two-dimensional parameter domain. Select the (41× 41 + 1)/2 = 841 sampling points to be used as hot core points (the red point in Figure 4 needs to be mapped to the three-dimensional space). Since the sampling points in the same row and column as the hot core point belong to the overlapping part of the two segments, the sampling points in the same row and column as the hot core point will be deleted. Solve the geodesic distance from the sampling point to the thermal core point in each surface segment $P_{i}$ after deleting the duplicate sampling points; sum the geodesic distances from the sampling point to the thermal core point in each segment; obtain the sum of the geodesic distances $d_{i}$ of each segment sampling point, and take it as the global fluctuation change index of the free-form surface. The sum of the geodesic distances $d_{i}$ of each segment $P_{i}$ is shown in Table 1.

As shown in Figure 5, each surface segment $P_{i}$ is further subdivided into four surface segments $P_{ij}$, and the sum of the geodesic distances $d_{ij}$ from the sampling point to the hot core point (red point in Figure 5) in each segment $P_{ij}$ is calculated. The results are shown in Table 2. Finally, the sum of the geodesic distances $d_{ij}$ of each free-form surface segment is obtained, which is used as an indicator of global fluctuation. Based on this indicator, the sampling quantity of each free-form surface segment is determined.

## 4. Sampling Strategy

In this section, firstly, the sampling quantity of each surface segment is calculated based on the sum of the geodesic distances of each surface segment solved in Section 3, and then the positions of the sampling points are generated based on the Isoparametric distribution, Poisson distribution, Hammersley distribution and NRook distribution. Finally, the geometric error evaluation method of the reconstructed surface and the original surface is determined.

### 4.1. Determine the Sampling Quantity

In this paper, a relative proportion method based on the sum of the geodesic distances of each segment is used to determine the P of each surface segment. The specific solution steps are as follows.

Step 1: According to the results of Section 3, calculate the average value of the sum of the geodesic distances of $P_{1}$, $P_{2}$, $P_{3}$ and $P_{4}$; obtain the ratio $\lambda_{i}$ of the sum of the geodesic distances of $P_{1}$, $P_{2}$, $P_{3}$ and $P_{4}$ and $d_{i}$. The reference sampling number $\bar{N_{i}}$ of each surface segment $P_{i}$ is the ratio of the total number of samples N and the number of segments φ. The sampling number of each surface segment $P_{i}$ is the product of the number of reference samples $\bar{N_{i}}$ and $\lambda_{i}$. The formula is as follows:(10)$$\lambda_{i}=\frac{d_{i}}{\bar{d}}$$
(11)$$\bar{N_{i}}=\frac{N}{\varphi }$$
(12)$$N_{i}=\bar{N_{i}}*\lambda_{i}$$

Step 2: Each surface segment $P_{i}$ continue to subdivide into surface segment $P_{ij}$. According to the principle of step 1, calculate the average value $\bar{d_{i}}$ of the sum of the geodesic distances of $P_{ij}$; obtain the ratio $\lambda_{ij}$ of the sum of the geodesic distances of $P_{i1}$, $P_{i2}$, $P_{i3}$ and $P_{i4}$ and $d_{ij}$. The reference sampling number $\bar{N_{ij}}$ of each surface segment $P_{i}$ is the ratio of the total number of samples $N_{i}$ and the number of segments φ; The sampling number of each surface segment $P_{ij}$ is the product of the number of reference samples $\bar{N_{ij}}$ and $\lambda_{ij}$. The sampling number of each surface segment $P_{ij}$ is the calculation result of step 1. The formula is as follows:(13)$$\lambda_{ij}=\frac{d_{ij}}{\bar{d_{i}}}$$
(14)$$\bar{N_{ij}}=\frac{N_{i}}{\varphi }$$
(15)$$N_{ij}=\bar{N_{ij}}*\lambda_{ij}$$

According to the requirements of measurement accuracy, the total number of sampling points of the free-form surface is determined to be 1600, φ = 4. Calculate the number of sampling points of each surface segment as shown in Table 3.

### 4.2. Determine the Sampling Position

After determining the number of sampling points in each segment, a specific point distribution algorithm is used to distribute the sampling points. In this paper, the sampling points generated based on the Isoparametric distribution, Poisson distribution, Hammersley distribution and NRook distribution are studied.
(a)Isoparametric distribution

Isoparametric distribution is to map the free-form surface from the three-dimensional space to the two-dimensional parameter domain, sample the two-dimensional parameter domain according to a certain step size to obtain the sampling points in the two-dimensional parameter domain, remap the sampling points in the two-dimensional parameter domain to the three-dimensional space, and obtain the set of sampling points in the three-dimensional space. It is calculated by Formulas (16) and (17):(16)$$u=u_{min}+\left(i-1\right)\frac{u_{max}-u_{min}}{\left(N_{u}-1\right)};i=1,2,\cdots ,N_{u}$$
(17)$$v=v_{min}+\left(i-1\right)\frac{v_{max}-v_{min}}{\left(N_{v}-1\right)};i=1,2,\cdots ,N_{v}$$
where $u_{min}$ is the minimum value of the *u* direction parameter; $u_{max}$ is the maximum value of the *u* direction parameter; $N_{u}$ is the number of sampling points in the *u* direction; $v_{min}$ is the minimum value of the *v* direction parameter; $v_{max}$ is the maximum value of the *v* direction parameter; $N_{v}$ is the number of sampling points in the *v* direction.
(b)Poisson distribution

Poisson distribution is a more uniform distribution mode compared to Isoparametric distribution. This distribution method can generate a random point set. The distribution method adopted is that every two points are at least on a specified minimum distance. The algorithm takes the range of the $R^{n}$ sample domain, the minimum distance r between samples and the constant k as the inputs. The steps are as follows [30]. 

Step 1: Initialize the n dimensional background grid, which is used to store variables and spatial search data. The search cell size is $r/\sqrt{n}$, and each grid cell contains, at most, one sample. Therefore, the grid is an n dimensional integer array. The default index (−1) indicates that there is no sample, and the non-negative integer indicates the index of the sample in the single grid.

Step 2: The initial sample $x_{0}$ is randomly selected from the sample field, inserted into the background grid, and this uses index (0) to initialize the activity list (sample index list).

Step 3: When the active list is not empty, a random index is selected from the list, and evenly selects k points from the spherical ring between the surrounding radius r  and the radius 2r. For each point, check whether it is within the radius r of the existing sample (use the background grid to test only the nearby samples). If a point is far enough from the existing sample, it is taken as the next sample and added to the existing index. If no such point is found after k attempts, the index is removed from the active list.
(c)Hammersley distribution

The Hammersley distribution is one of the most prominent uniform distribution sampling algorithms at present. This method is based on the computer binary number representation method, which converts a given decimal number into a binary number, inverts the order, multiplies each number on each bit of the binary by a power series with 1/2 as the base number and the corresponding number of digits as the exponent, and cumulatively sums them. The calculation result is placed after the decimal point to form the sampling value. The sampling formula is as follows:(18)$$u_{i}=i/N$$
(19)$$v_{i}=\sum_{j=0}^{k-1}b_{ij}2^{-j-1}$$
(20)$$k=\left[\log_{2}N\right]$$
where N is the number of sampling points; i is the ith sampling point, and its range is $\left[0,\;N-1\right]$; $b_{i}$ is the binary representation of the index; $b_{ij}$ is the jth bit representing $b_{i}$, and the range of j is $\left[0,\;k-1\right]$; k is b number of digits of $b_{i}$.
(d)NRook distribution

NRook distribution is a uniform distribution algorithm based on the principle of a chess board. If a square is divided into n∗n small squares and n sampling points are placed inside, there is only one sampling point in each row and column. The algorithm steps are as follows. 

Step 1: Initialize the sampling set Q and the sampling step λ according to the number of samples N. The calculation formula is as follows:(21)$$Q=\left[0,\;1,\cdots ,N-1\right]$$
(22) λ=1/N

Step 2: Selecting random samples $n_{i}$ from the sample set Q; select the random number $\;p_{i}$, $q_{i}$ from [0, 1], $u_{i}$ is the sum of the result of multiplying the sampling step λ by $n_{i}$ and a random number $p_{i}$, $v_{i}$ is the sum of the result of multiplying the sampling step λ by the sampling times i and a random number $q_{i}$. The calculation formula is as follows:(23)$$\left\{\begin{array}{c} u_{i}=\lambda n_{i}+p_{i}\;\\ v_{i}=\lambda i+q_{i} \end{array}\right.$$

According to the basic principles of the four distribution modes of Isoparametric distribution, Poisson distribution, Hammersley distribution and NRook distribution, the distribution sequence of parameter $u\in \left[0,1\right]$ and parameter $v\in \left[0,\;1\right]$ is calculated and generated. The number of samples is set to 700. The results are shown in Figure 6. 

The sampling points generated by the four distribution methods, i.e., Isoparametric distribution, Poisson distribution, Hammersley distribution and NRook distribution, need to be remapped to the corresponding sections according to the surface segment interval, and the sampling points are reconstructed. This is described in detail in Section 5.

### 4.3. Error Comparison Method between the Reconstructed Surface and Original Surface

The reconstructed surface and the original surface after surface reconstruction need to be evaluated for the geometric error of reconstruction. The method adopted in this paper is shown in Figure 7. The Isoparametric method is used to take a series of sampling points on the original surface, and make a straight line perpendicular to the z axis of the oversampling point. The straight line has an intersection with the reconstructed surface. The x and y coordinate values of the intersection and the sampling point are the same. The difference between the z coordinate values of the sampling point and the intersection is calculated, which is the reconstructed geometric accuracy of the reconstructed surface and the original surface.

By solving the z coordinate value difference between each sampling point on the original surface and the intersection point on the reconstructed surface, the root mean square error (RMSE) and the global maximum error (ME) are used as the reconstruction accuracy indexes of the reconstructed surface and the original surface. The formula is as follows:(24)$$\mathrm{MSE}=\frac{1}{N}\overset{N}{\underset{i=1}{\sum }}{\left\|z_{i}-{z_{i}}^{\prime }\right\|}^{2}$$
(25)$$\mathrm{RMSE}=\sqrt{\frac{1}{N}\overset{N}{\underset{i=1}{\sum }}{\left\|z_{i}-{z_{i}}^{\prime }\right\|}^{2}}\;$$
(26)$$\mathrm{ME}=\max\left|z_{i}-{z_{i}}^{\prime }\right|\;$$
where N is the number of sampling points; $z_{i}$ is the *z* coordinate value of the original surface sampling point; ${z_{i}}^{\prime }$ is the coordinate value of the reconstructed surface sampling point.

## 5. Experiment and Discussion

The free-form surface is reconstructed according to the different sampling strategies. The reconstructed surface is analyzed based on the reconstruction accuracy indexes, RMSE and ME, of the reconstructed surface and the original surface to evaluate the impact of the different sampling strategies on the reconstruction’s accuracy.

The parameter interval obtained by the four distribution methods of Isoparametric distribution, Poisson distribution, Hammersley distribution and NRook distribution is $\left[0,\;1\right]$, and the length of the parameter u and v interval of the surface segment is 0.25. Therefore, it is necessary to map the boundary range of the divided surface to the corresponding interval. The interval range of each surface segment is shown in Figure 8.

According to the sampling number of each surface segment in Figure 8, four distribution methods, i.e., Isoparametric distribution, Poisson distribution, Hammersley distribution and NRook distribution, are used to calculate and generate sampling points, and map them to the corresponding interval range of each surface segment in Figure 8. The original free-form surface sampling point set obtained is used for the free-form surface reconstruction. The reconstructed surface is shown in Figure 9, where the green point represents the sample point, and the brown surface represents the reconstructed surface. Different sampling methods produce a different distribution of the sample points, and the reconstructed surface is also different.

The reconstructed surface is compared with the original surface in terms of the reconstructed geometric error, and the ME of each point $\left(x,y,z\right)$ on the reconstructed surface is converted into the corresponding color. The obtained error distribution results are shown in Figure 10. Figure 10a is the reconstructed geometric error without surface blocking and other parameter distribution; Figure 10b is the reconstructed geometric error of Poisson distribution; Figure 10c is the reconstructed geometric error of Hammersley distribution; and Figure 10d is the reconstructed geometric error of NRook distribution.

Table 4 and Figure 11 shows the error comparison results between the reconstructed surface obtained by surface reconstruction and the original surface after the sampling points of the free-form surface are obtained by the different distribution methods. Comparing Isoparametric distribution from the aspect of reconstruction ME, the reconstruction error of the hybrid adaptive sampling method, based on geodesic distance combined with Poisson distribution, Hammersley distribution and NRook distribution, is reduced by 47.12%, 41.56% and 20.22%, respectively, compared with that of the Isoparametric distribution sampling method. Compared with the adaptive sampling method proposed in Reference [24], the reconstruction error of the hybrid adaptive sampling method based on the geodesic distance combined with Poisson distribution, Hammersley distribution and NRook distribution, is reduced by 69.83%, 66.67% and 54.49%, respectively. The reconstruction error of the adaptive hybrid sampling method, based on the geodesic distance combined with Poisson distribution, Hammersley distribution, NRook distribution and the adaptive sampling method used in Reference [14], is reduced by 62.46%, 62.15% and 52.61%, respectively, compared with Poisson distribution, Hammersley distribution and NRook distribution. 

The adaptive hybrid sampling method based on geodesic distance significantly reduced the reconstruction ME compared with the general Isoparametric distribution sampling method and the adaptive sampling method in Reference [24], and the reconstruction accuracy has been greatly improved. The reconstructed ME based on the adaptive hybrid sampling method of geodesic distance is significantly improved compared with the sampling methods of Poisson distribution, Hammersley distribution and NRook distribution that are not based on the adaptive hybrid sampling method of geodesic distance, which verifies the effectiveness of the adaptive hybrid sampling method based on geodesic distance in reducing the reconstructed ME.

From the aspect of reconstruction RMSE, the adaptive hybrid sampling method based on geodesic distance combined with Poisson distribution, Hammersley distribution and NRook distribution, reduces the reconstruction RMSE by 24.06%, 28.95% and 20.00%, respectively, compared with the sampling method of Isoparametric distribution. Compared with the adaptive sampling method proposed in Reference [24], the reconstructed RMSE of the hybrid adaptive sampling method based on geodesic distance combined with Poisson distribution, Hamersley distribution and NRook distribution is reduced by 58.44%, 61.11% and 56.17%, respectively. The reconstructed RMSE of the adaptive hybrid sampling method based on geodesic distance combined with Poisson distribution, Hammersley distribution and NRook distribution is reduced by 56.75%, 73.31% and 82.63%, respectively, compared with the sampling methods of Poisson distribution, Hammersley distribution and NRook distribution. 

The RMSE reconstruction of the adaptive hybrid sampling method based on the geodesic distance combined with Poisson distribution, Hammersley distribution and NRook distribution is greatly reduced compared with the sampling method of Isoparametric distribution and the adaptive sampling method in Reference [24], and the reconstruction accuracy is significantly improved. The RMSE of the adaptive hybrid sampling method based on geodesic distance combined with Poisson distribution, Hammersley distribution and NRook distribution is significantly reduced compared with the sampling method based on Poisson distribution, Hammersley distribution and NRook distribution without geodesic distance, and the reconstruction accuracy is greatly improved.

In order to further verify the effectiveness of the proposed method, this paper uses an ABB IRB1200 robot and Creaform MetraSCAN-R BLACK for experimental verification. The measurement accuracy is 25 µm, and the measurement depth is 250 mm, as shown in Figure 12.

Set the number of sample points to 100, and use the self-developed path algorithm to obtain the robot scanning measurement path, as shown in Figure 13. The free-form surface is scanned and measured. The measured results are shown in Figure 14. 

In order to further verify the effectiveness of the sampling algorithm, repeated measurement experiments are carried out. Taking the first measurement data as the error judgment standard, analyze the error of the two measurement experiments. The error results are shown in Figure 15. The average deviation is 0.001 mm and the standard error is 0.013 mm, which verifies the effectiveness and robustness of the sampling method proposed in this paper.

## 6. Conclusions

In this paper, an adaptive hybrid sampling method for a free-form surface based on geodesic distance is proposed, which can be used for the precise measurement of geometric parameters of free-form surfaces. This method can effectively improve measurement efficiency.

The geodesic distance is introduced as a measure of the global fluctuation of the free-form surface. The free-form surface is divided into multiple surface segments. The sampling number of each surface segment is determined according to the sum of the geodesic distances of each surface segment within the surface segment. The sampling points of each surface segment are generated based on the Poisson distribution, Hammersley distribution and NRook distribution sampling methods. Finally, the sampling points of the overall free-form surface are obtained.

The reconstruction error of the free-form surface obtained by this method was compared with the common sampling strategies. The results show that the adaptive hybrid sampling method of a free-form surface based on geodesic distance can effectively reduce the reconstruction error and significantly improve the reconstruction accuracy of free-form surfaces.

In future work, the sampling distribution method can be self-optimized by combining intelligent algorithms, to further improve the reconstruction accuracy of free-form surfaces.

## Figures and Tables

**Figure 1 sensors-23-03224-f001:**
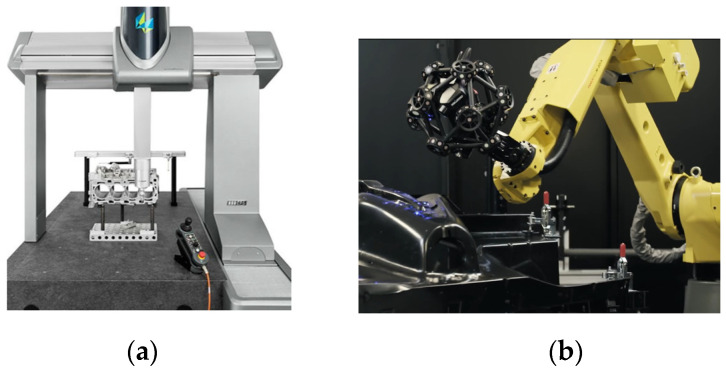
Two measurement methods. (**a**) CMM; (**b**) Robot optical scanning system.

**Figure 2 sensors-23-03224-f002:**
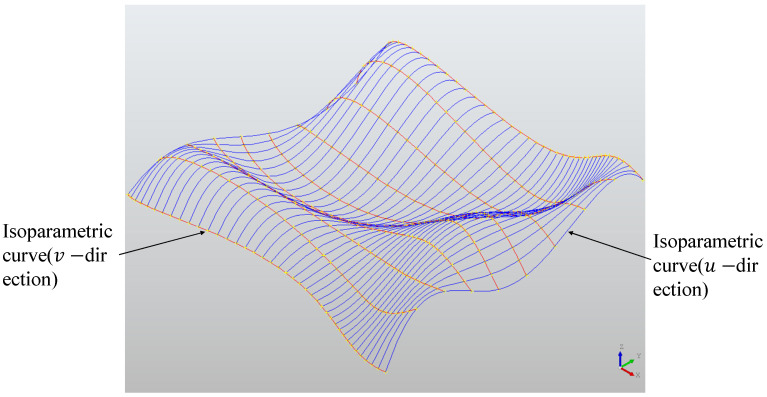
NURBS surface and its Isoparametric mesh.

**Figure 3 sensors-23-03224-f003:**
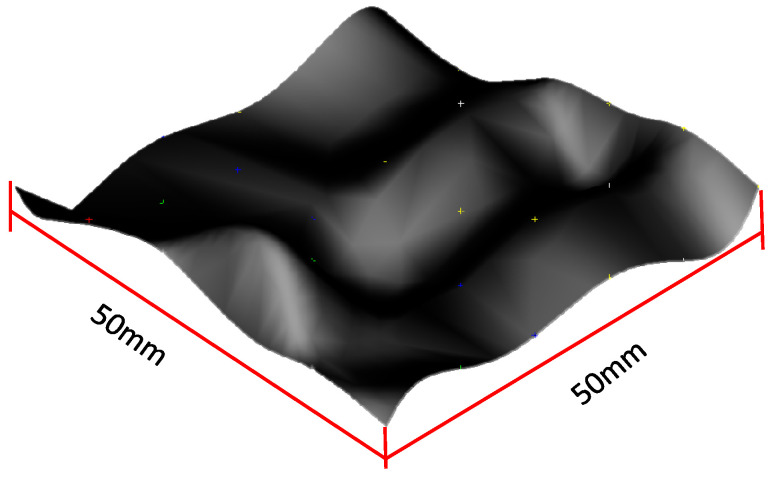
Free-form surface.

**Figure 4 sensors-23-03224-f004:**
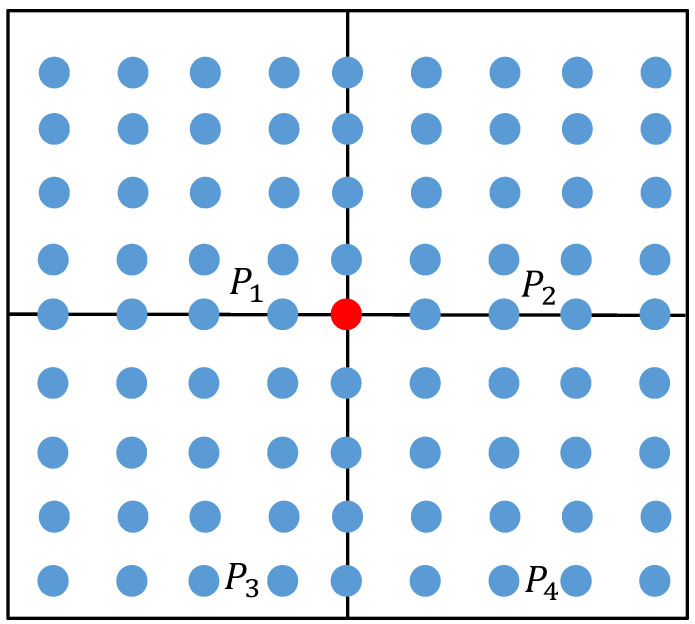
Schematic diagram of the geodesic distance of the first surface segment.

**Figure 5 sensors-23-03224-f005:**
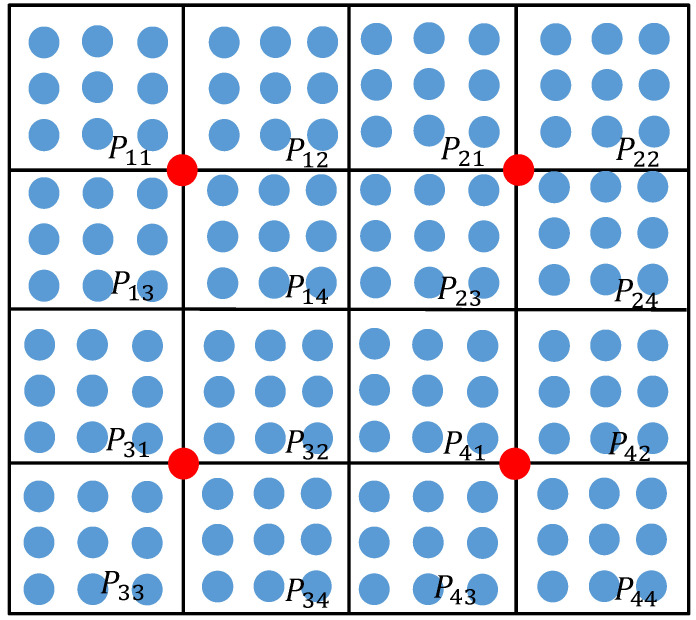
Schematic diagram of the segment geodesic distance of the second surface.

**Figure 6 sensors-23-03224-f006:**
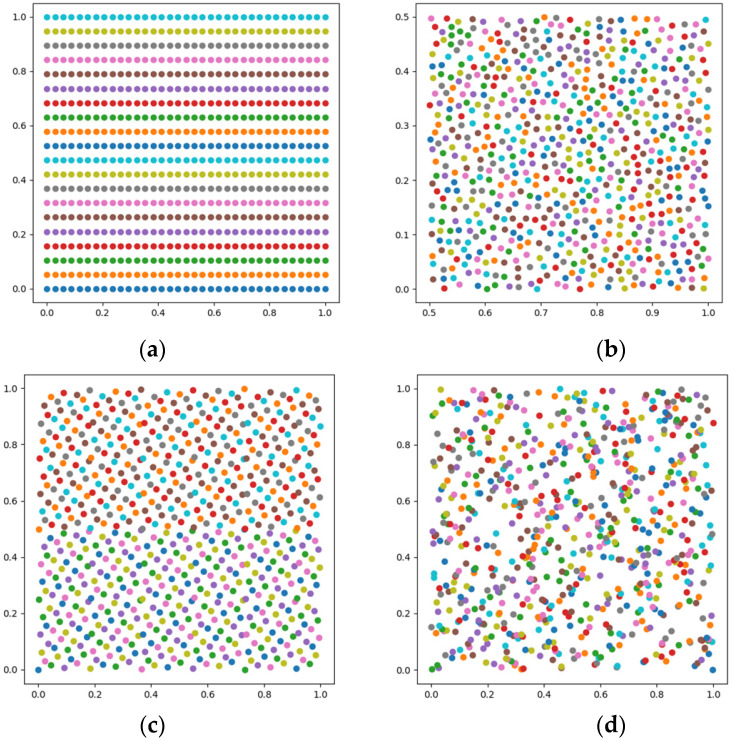
Sampling distribution (**a**) Isoparametric distribution; (**b**) Poisson distribution; (**c**) Hammersley distribution; (**d**) NRook distribution.

**Figure 7 sensors-23-03224-f007:**
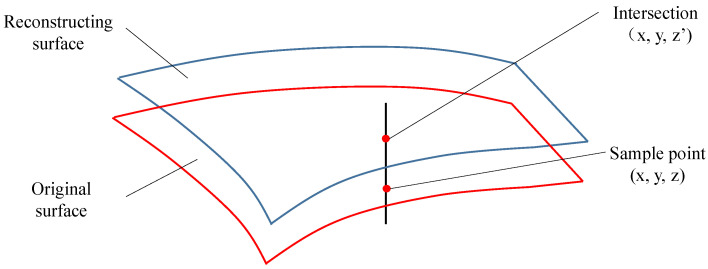
Schematic diagram of the reconstruction error surface.

**Figure 8 sensors-23-03224-f008:**
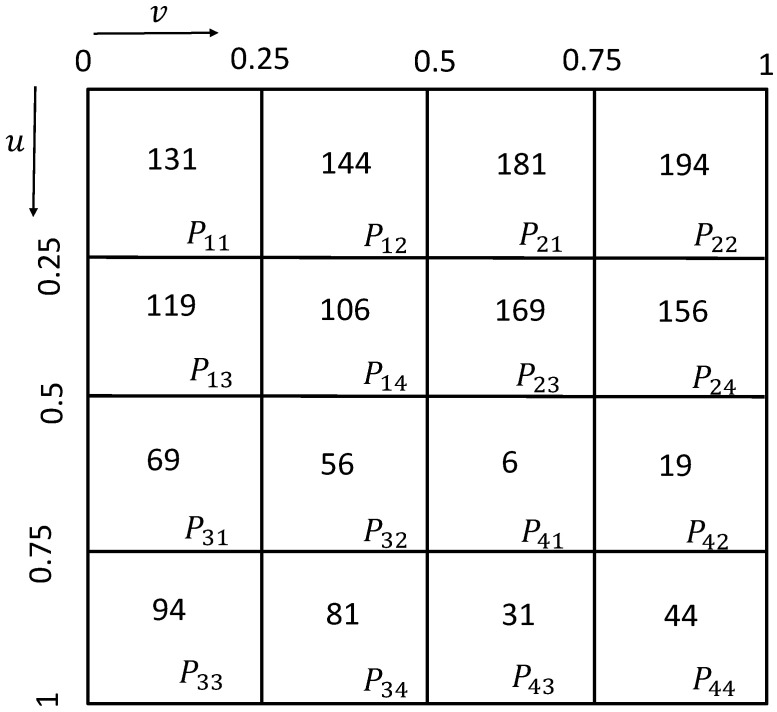
Interval range and sampling quantity of the surface segment.

**Figure 9 sensors-23-03224-f009:**
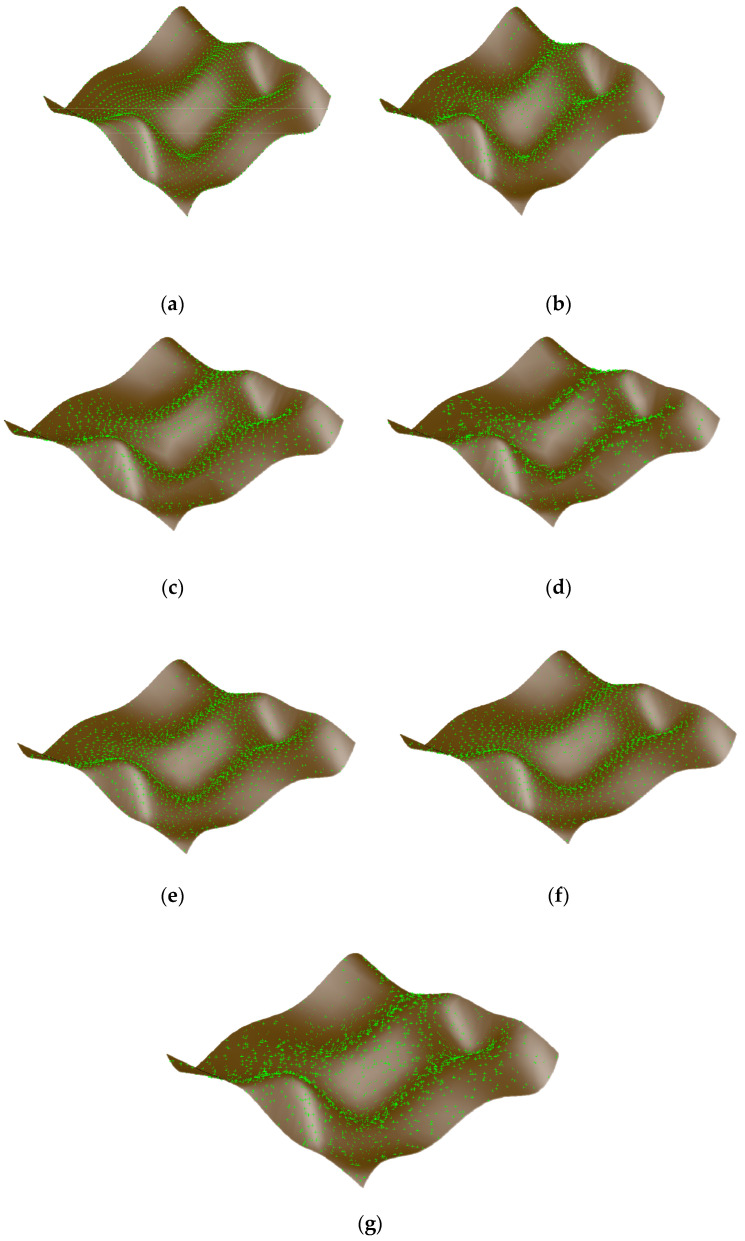
Sampling point distribution of the free-form surface and surface reconstruction: (**a**) Isoparametric; (**b**) Poisson; (**c**) Hammersley; (**d**) NRook; (**e**) Adaptive hybrid Poisson; (**f**) Adaptive hybrid Hammersley; (**g**) Adaptive hybrid NRook.

**Figure 10 sensors-23-03224-f010:**
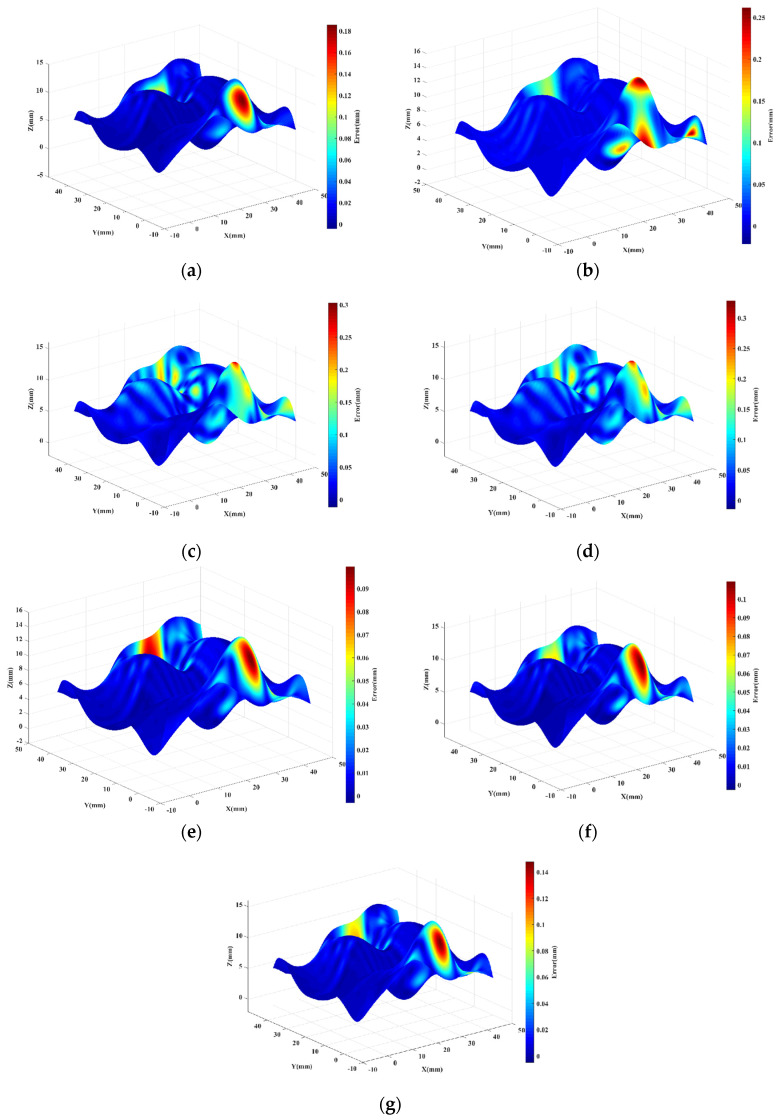
Reconstructed surface ME distribution map: (**a**) Isoparametric; (**b**) Poissson; (**c**) Hammersley; (**d**) NRook; (**e**) Adaptive hybrid Poisson; (**f**) Adaptive hybrid Hammersley; (**g**) Adaptive hybrid NRook.

**Figure 11 sensors-23-03224-f011:**
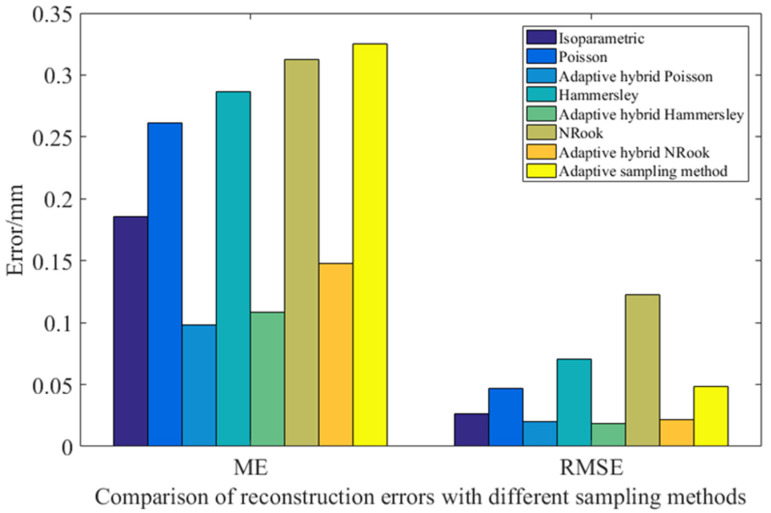
Comparison of the reconstruction errors using different sampling methods.

**Figure 12 sensors-23-03224-f012:**
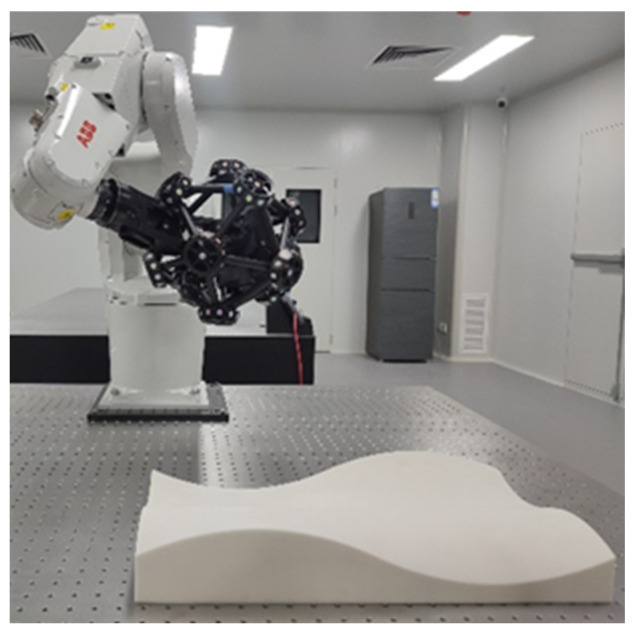
Measuring the surface using a MetraSCAN laser scanner.

**Figure 13 sensors-23-03224-f013:**
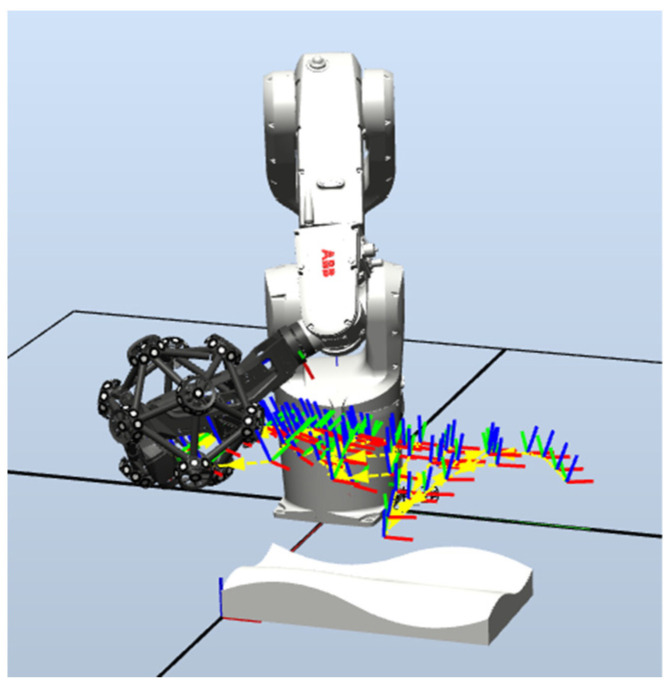
Robot scanning measurement path.

**Figure 14 sensors-23-03224-f014:**
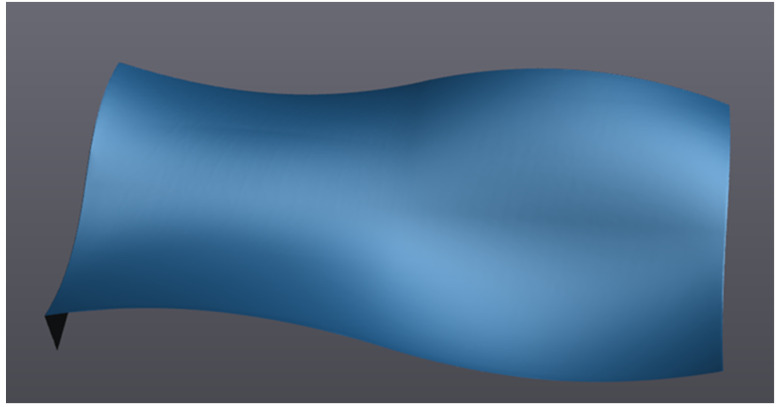
Point cloud reconstruction results of the free-form surface.

**Figure 15 sensors-23-03224-f015:**
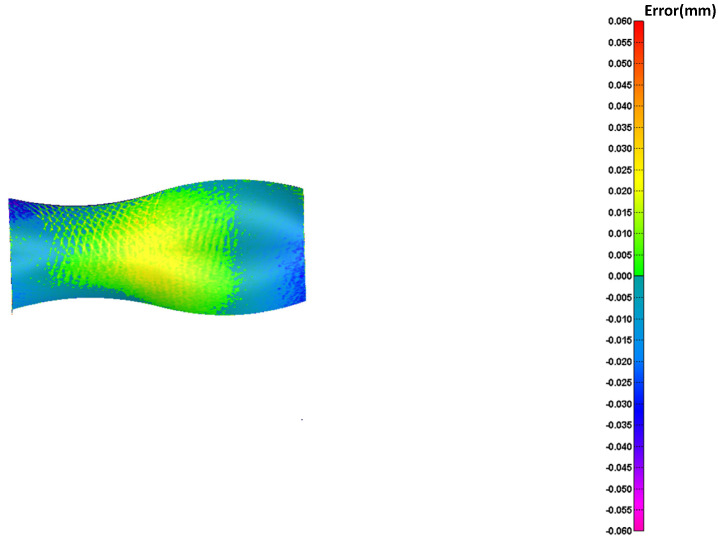
Error comparison results.

**Table 1 sensors-23-03224-t001:** Sum of the geodesic distances of the first surface segment.

Segment	Distance/mm
1	7750
2	7816
3	7578
4	7171

**Table 2 sensors-23-03224-t002:** Sum of the geodesic distances of the second quadric surface segment.

Segment	Distance/mm
11	4207
12	4305
13	3983
14	3714
21	4174
22	4186
23	3824
24	3728
31	3546
32	3442
33	4059
34	4004
41	3366
42	3427
43	4069
44	4163

**Table 3 sensors-23-03224-t003:** Number of sampling points for each surface segment.

Segment	Sample Size
11	131
12	144
13	119
14	106
21	181
22	194
23	169
24	156
31	69
32	56
33	94
34	81
41	6
42	19
43	31
44	44

**Table 4 sensors-23-03224-t004:** Comparison of the reconstruction errors using different sampling methods.

Sampling Method	ME/mm	RMSE/mm
Isoparametric	0.1855	0.0266
Poisson	0.2613	0.0467
Adaptive hybrid Poisson	0.0981	0.0202
Hammersley	0.2864	0.0708
Adaptive hybrid Hammersley	0.1084	0.0189
NRook	0.3123	0.1226
Adaptive hybrid NRook	0.1480	0.0213
Adaptive sampling method [24]	0.3252	0.0486

## Data Availability

Data available on request due to restrictions, e.g., privacy or ethical.

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
