# Peer review of "An Adaptive Hybrid Sampling Method for Free-Form Surfaces Based on Geodesic Distance"

_sensors, 2023, doi:10.3390/s23063224_

Round 1

Reviewer 1 Report

The work is written in correct language and contains most of the important information. However, it has not shied away from a few underdevelopments. I attach the comments below.

At the outset, please very clearly define the purpose and motivations of the action taken. Currently, this form is unsatisfactory.

There have also been some minor omissions at work.

I would ask that equation 1 line 161-165 be more clearly separated. Currently the text blends with the legend and in my opinion it is very unreadable. Please apply this remark to all equations and descriptions. Please review the text again.

Line 173 ":" please put on one line with the sentence from line 172.

Line 196 no space between references to literature [29].

I would very much like to ask you to add a description to figure 2. A word of commentary will definitely ennoble the work. As it stands now, it is not sufficient.

Please center equations 4-7 and others in the text.

Step 1: Solve the thermal kinematics equation (5) describing the propagation state of 197

heat, and establish the temperature scalar field U. The time dispersion of the thermal prop-198

agation equation is.

My question is about field U. Are you sure the authors have labeled this size correctly?

I recommend that Tab. 1 be placed directly below the text. And the description of the information is in the passage 223-236 is in my opinion imprecise and a bit confusing. I would very much ask that this passage be edited in a more precise way.

Table 2 in my opinion would fit much better in front of illustrations 4 and 5. I would suggest to consider this kind of change in the text.

Instead of Tables 1 and 2 under Figure 5, please provide a word of commentary/summary.

Another editing note, Table 3 is badly formatted and misaligned. Please use the same style and alignment throughout the text, tables, figures and designs to the center of the sheet.

I would ask under Figure 9 for a word of commentary and description of the drawings, what is important here, why the authors presented so many variations, what makes these drawings different, etc. I would also very much appreciate it if you could point out in the selected drawing what the reader should pay attention to. Description is required here.

In the excerpt from lines 403-409, please use the correct nomenclature for figure references, e.g. Fig. 10a, Fig. 10b, etc.

In this case, it is mandatory to add a comment under the figures explaining the essence of the data presented. As before, I would place Table 4 in a different place, however, I will leave this issue to the Authors.

Under tab 4 add a space.

I also recommend adding a simple chart to illustrate the data in the table. This will definitely improve the readability of the results.

Author Response

We appreciate the reviewer for this kind recommendation. We have carefully studied all the opinions and carefully corrected the mistakes. The main corrections in the paper and the replies - comments to the reviewers are marked in red in the attachment. Best wishes to you.

Reviewer 2 Report

This paper proposes an adaptive hybrid sampling method for free-form surfaces based on geodesic distance. As a user of 3D detection, it can be said that this starting point is very good, very worthy of study. Especially for non-contact profilometer and CMM, the correct selection of sampling point distribution has very important practical significance. However, there are problems in this article. Therefore, I give my opinion (major revision).

First of all, this paper lacks detailed physical experiments and data analysis, which greatly reduces the reliability of the discussion. It is the most basic and necessary to make corresponding algorithm comparison for the actual measurement of a free-form surface or even an aspheric surface. However, in this article, it is missing.

Secondly, the language of this article needs further polishing. Not only that, there are several obvious errors in the article. For example, the author name of 'crane' in line 190 and the image annotation in lines 227 and 239.

Thirdly, in the simulation experiment of this paper, in order to prove the effectiveness of the new algorithm proposed in this paper, the algorithms used for comparison are all based on blind sampling, without combining the characteristics of the free-form surface itself. Can it be compared with the algorithm in citations 13-25 in your paper? Compared with advanced algorithms, paper innovation can be highlighted.

In addition to the above comments, there are a number of other issues as follows:

1. In the process of surface reconstruction, which method should be used for surface reconstruction after the location of sampling points is allocated?

2. In 4.3, when reconstruction error evaluation is carried out, isoparametric sampling method is used to assign the locations of sampling points, and whether there is a lack of randomness ?

Author Response

(The authors gave the same response as above.)

Round 2

Reviewer 2 Report

Some work should also be done on language polishing, and the use of isoparameans is conducive to the algorithm presented in this paper, and more random methods should be used as validation.
